# Rare earth metals production using alternative feedstock that eliminates HF

Anirudha Karati[1], Harshida Parmar[1], Trevor Riedemann[2], Matthew Besser[2], Denis Prodius[1,3] & Ikenna C. Nlebedim [1,3] ✉

This work reports the successful production of rare earth (RE) metal using Na-RE-F. Presently, RE metals are primarily produced using RE-fluoride due to its higher air and moisture stability compared to RE-chloride. However, its preparation requires the use of corrosive and hazardous chemicals, such as hydrofluoric acid (HF) or ammonium bifluoride ($NH_4HF_2$). The present study demonstrates that Na-RE-F is an alternative salt to the typically used RE-fluoride. The Na-RE-F for this work is produced via a scalable hydrometallurgical approach using three different RE salts as feedstock, including acetate, nitrate, and chloride. HF is neither used nor generated during the salt preparation process. Furthermore, the Na-RE-F powder dries in air (without dry HF), and only water evolves during the drying process. Analyses of the Na-RE-F show that NaF liberates as a flux during the heating process, which lowers the salt reduction temperature to <900 °C, thus minimizing or eliminating the need for additional flux. Calciothermic reduction of the Na-RE-F salt is successfully employed to obtain RE metal. This work represents a safer, greener, and more widely deployable approach for producing the RE metals needed for permanent magnets which support the transition to a cleaner society through the decarbonization of the transportation industry.

Rare earth elements (REEs) are increasingly susceptible to supply risks due to their limited geographical availability and the fact that they are subject to rising geopolitical tensions. Some REEs, such as neodymium (Nd), praseodymium (Pr), dysprosium (Dy), and terbium (Tb), are used for permanent magnets that support many clean energy and renewable energy applications, e.g., electric vehicles, defense systems, electronics, etc. Hence, REEs have been termed "the vitamins of modern industry"[1–5].

Reduction of rare earth (RE) salts to metals is typically accomplished through molten salt electrolytic or metallothermic methods. The molten salt electrolytic method is the widely used approach because of its relatively simple process-flow and better suitability for continuous production[6]. The electrolyte for the molten-salt bath can be chloride- or fluoride-based, and the feedstock can be RE-chloride or RE-oxides, respectively. The molten salts for both the chloride and fluoride electrolytic routes require heating a mixture of RE-halides and

alkali or alkaline earth salts to 800–1000 °C. Unlike the metallothermic methods, the electrolytic methods use consumable graphite electrodes which results in $CO_2$ emissions[7–11]. Fluoride-based baths have higher current efficiency (<87%), compared to chloride-based baths (<50%)[12] because the hygroscopic nature of RE-chlorides and the formation of oxychlorides limit yield and contaminate the obtained RE-metal product.

In the metallothermic methods, the RE-feedstock materials are exothermically reduced to metals using more reactive metals like calcium (calciothermic reduction)[13]. This method can use RE-oxides, chlorides, or fluorides as feedstock. RE-fluoride is less hygroscopic than RE-chloride and both have lower melting temperatures than RE-oxide. The energy cost for direct metallothermic reduction of RE-oxides, as well as the potential for oxidation of the obtained RE-metals due to the lack of a protective flux, makes it less practical. Although calciothermic reduction of RE-chloride has been attempted, it was

[1]Division of Critical Materials, Ames National Laboratory, Ames, IA, USA. [2]Division of Materials Science and Engineering, The Materials Preparation Center, Ames National Laboratory, Ames, IA, USA. [3]These authors contributed equally: Denis Prodius, Ikenna C. Nlebedim. ✉e-mail: nlebedim@ameslab.gov

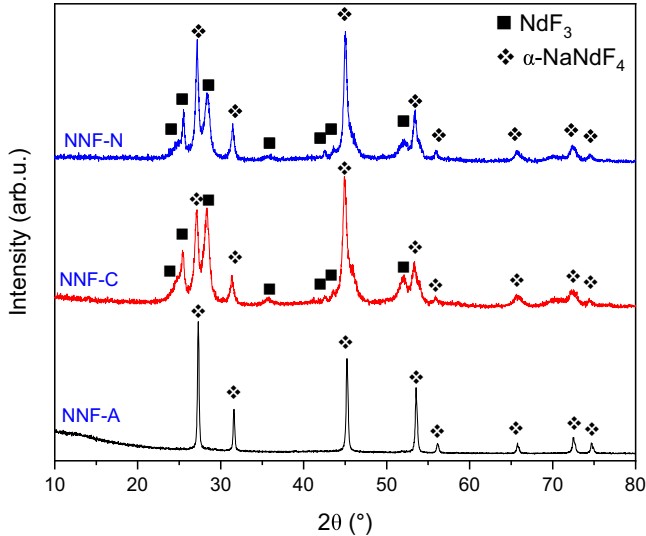

**Fig. 1 | X-ray diffraction (XRD) patterns of the samples synthesized with the three precursor salts.** The diffraction pattern for the sample synthesized using the acetate precursor, i.e., NNF-A, comprised of a single α-NaNdF$_4$ phase (denoted with a diamond symbol). The NNF-C and NNF-N samples (synthesized using the chloride and nitrate precursors, respectively) had two phases corresponding to α-NaNdF$_4$ and NdF$_3$ (denoted with a solid square symbol) phases.

**Table 1 | Measured fractions of α-NaNdF$_4$ and secondary NdF$_3$ phases contained in the as-synthesized fluoride salts from the various precursor materials**

| As-synthesized fluor-ide salt | Weight % of phases in as-synthesized material | |
|---|---|---|
| | NdF$_3$ | α-NaNdF$_4$ |
| NNF-A | - | 100 |
| NNF-C | 10 ± 2 | 90 ± 2 |
| NNF-N | 6 ± 1 | 94 ± 1 |

limited by low yield, and the difficulties with handling the hygroscopic and high vapor pressure RE-chlorides[14].

From the foregoing discussion, the practical (hence preferred) application of both the electrolytic and metallothermic approaches requires RE-fluoride, which can be prepared via wet or dry processing techniques. The former involves the wet chemical reaction of RE chloride or nitrate salts and hydrofluoric acid (HF), followed by drying at 600–700 °C in a dry hydrogen fluoride environment[15]. In the dry processing method, RE$_2$O$_3$ is directly reacted with an excess of dry hydrogen fluoride (200%) at ≥700 °C to obtain anhydrous RE-fluoride[16,17]. The health hazards[18–24] associated with using or generating HF create a considerable technical gap in the anhydrous RE-fluorides preparation for RE metal supply chain. Furthermore, the high melting point of RE-fluorides (>1300 °C) necessitates fluxing with LiF, NaF, KF, etc. during RE metal production. Thus, the need for an alternate feedstock to generate RE metals is timely.

In this work, we report the use of NaNdF$_4$ (NNF) to overcome most of the challenges faced in the RE metal production using the calciothermic reduction process. NNF has the following advantages (a) in-situ formation of NaF as a fluxing agent during reduction, (b) lower liquidus temperatures (~1100 °C)[25], (c) facile room temperature synthesis by co-precipitation[26], and (d) no requirement for toxic HF or NH$_4$HF$_2$. Furthermore, reports on using NNF in human cancer pathological studies indicate that it would be a non-toxic feedstock[27,28]. We have demonstrated the use of NNF for Nd metal production using the calciothermic process. Unlike conventional RE-fluoride reduction, NNF

not only ensures fluoride ion availability but also represents a strategic shift toward safer, more sustainable, and regulation-compliant rare earth metal production.

## Results

The reaction between Nd(CH$_3$COO)$_3$·2H$_2$O (neodymium acetate hydrate) and aqueous sodium fluoride solution yields single phase precipitate of α-NaNdF$_4$ (α-NNF) [space group: $Fm\bar{3}m$ (no. 225)] (Fig. 1), in agreement with a previous report [Powder Diffraction File (PDF) No. 00-028-1114]. The reaction proceeds as:

$$Nd(CH_3COO)_3 \cdot 2H_2O + 4NaF \rightarrow NaNdF_4 \cdot 0.3H_2O + 3CH_3COONa + 1.7H_2O$$

$$(1)$$

The reaction of NdCl$_3$·6H$_2$O (neodymium chloride hydrate) and Nd(NO$_3$)$_3$·6H$_2$O (neodymium nitrate hydrate) with sodium fluoride both yielded two phases (Fig. 1): α-NNF [space group: $Fm3m$ (no. 225)] and NdF$_3$ [space group: $P6_3/mcm$ (no. 193)] [Powder Diffraction File(PDF) No. 00-009-0416]. The contents of the NdF$_3$ phases were 10% for neodymium chloride hydrate and 5% for neodymium nitrate hydrate (Table 1). The formation of two phases in the products can be attributed to having more than one phase of the salts in the starting materials. For example, both NdCl$_3$ and NdOCl would be present for the chloride salt, as further explained in the supplementary information. A recent study by Gibson et al. reported a negative Gibbs' energy for NNF, indicating its thermodynamic stability at 298.15 K[29]. Additional phase analysis information is provided in Supplementary Figs. 2–4 and Supplementary Tables 1–3. The scanning electron microscopy (SEM) analysis of the fluoride samples in the present study was used to confirm that they have particle sizes of <100 nm (Supplementary Figs. 5–7). Hereinafter, the NaNdF$_4$ produced from acetate, chloride and nitrate will be referred to as NNF-A, NNF-C, and NNF-N, respectively.

The differential scanning calorimetry (DSC) data (Fig. 2a) depicts one exothermic and two endothermic peaks for all three samples. The NNF-A and NNF-N exhibit the first exothermic peaks at T = 394 °C and T = 355 °C, respectively. The first exothermic peak is not pronounced for NNF-C. This first peak for all the samples is a concomitant occurrence of two thermal events, namely, the volatilization of the water of crystallization (endothermic) and the conversion of α-NaNdF$_4$ to β-NaNdF$_4$ phase (exothermic)[26], with the latter being more prominent.

The endothermic peaks at 642 °C, 750 °C and 700 °C (broad peak) represent the melting of the eutectic mixture comprising NaF and β-NaNdF$_4$[25] for NNF-N, NNF-C, and NNF-A, respectively. The variation in the peak positions, and the broadness of the peak for NNF-A, are possibly due to the differing contents of the β-NaNdF$_4$ in the three fluorides. Also, respectively, the endothermic peaks at 815 °C, 813 °C, and 821 °C represent the β − NaNdF4 → α − NaNdF4 phase transformation in the samples[25,26].

The thermal gravimetric analysis (TGA) results show less than 2% mass loss for all the samples, with the NNF-A sample losing the most (Fig. 2b). This is possibly due to the higher phase fraction of NaNdF$_4$ in the NNF-A sample which, in turn, contributed to higher water of crystallization content. The TGA profile of NNF-A depicts two mass loss regions: (a) 150 °C to 300 °C for volatilization of adsorbed water and (b) >350 °C for volatilization of water of crystallization.

Mass spectrometric analysis was performed in the quasi-multiple ion detection (QMID) mode during the TG-DSC experiments and are depicted in Fig. 3.

The peaks from the QMID plots help determine the chemical species with a certain mass number that evolves during thermal treatment. There are two notable peaks of mass number 18 (due to water) in all three fluorides. The first peak at <300 °C represents the

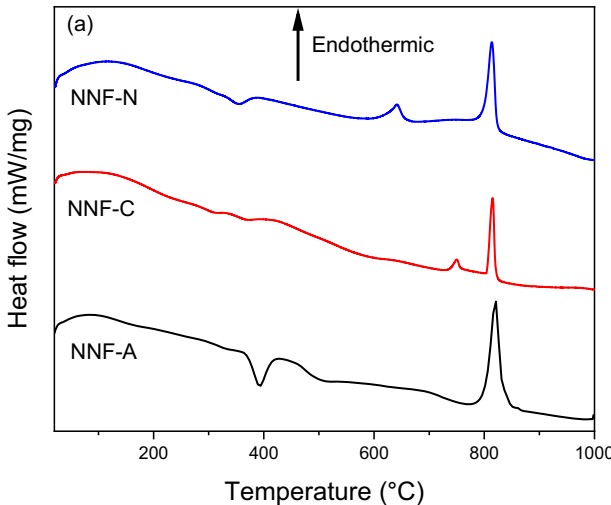

**Fig. 2 | Thermal analysis of the sodium neodymium fluoride samples obtained from neodymium acetate (NNF-A), neodymium chloride (NNF-C), and neodymium nitrate (NNF-N) precursors. a** The differential scanning calorimetry (DSC) plot of the three fluoride samples. The exothermic peak is due to $\alpha - NaNdF4 \rightarrow \beta - NaNdF4$ transformation, while the two endothermic peaks correspond to the melting of $\beta - NaNdF4$ and $\beta - NaNdF4 \rightarrow \alpha - NaNdF4$ transformation. **b** thermogravimetric analysis (TGA) patterns of the fluoride samples obtained from the different precursor salts. The higher mass loss for NNF-A is due to the higher $\alpha$-NaNdF$_4$ content.

**Fig. 3 | Mass spectrometric analysis of the sodium neodymium fluoride samples performed in quasi-multiple ion detection (QMID) mode. a** There are two peaks in the NNF-A sample corresponding to water at 220 °C and 400 °C, **b** The NNF-C has a sharp peak at ~100 °C and two less prominent peaks at 300 °C and 400 °C, and **c** The NNF-N has a sharp peak at ~100 °C and another prominent peak at 470 °C. All three fluorides did not exhibit the evolution of any hydrogen fluoride. NNF-A sample synthesized using the acetate precursor, NNF-C sample synthesized using the chloride precursor, NNF-N sample synthesized using the nitrate precursor.

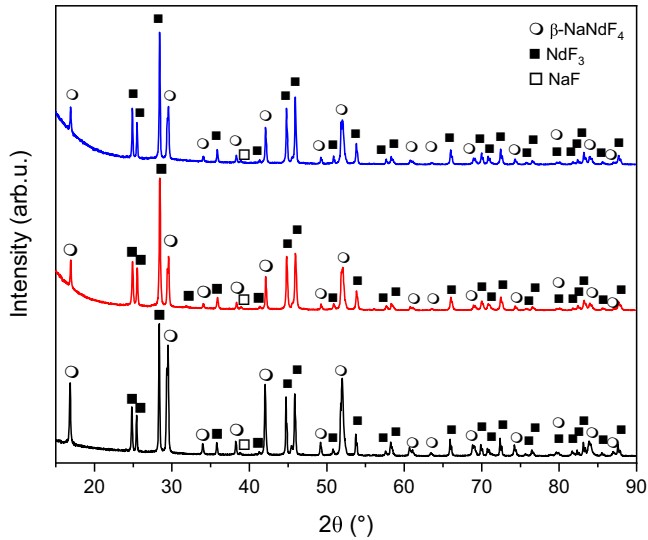

**Fig. 4 | XRD patterns obtained after drying the as-synthesized samples at 500 °C for 3 h.** The X-ray diffraction (XRD) pattern obtained by heating as-synthesized powders of NNF-A (black plot), NNF-C (red plot), and NNF-N (blue plot) at 500 °C for 3 h shows three phases for each of the samples. The open circle, solid square, and open square symbols correspond to β-NaNdF$_4$, NdF$_3$, and NaF phases, respectively. NNF-A, sample synthesized using the acetate precursor; NNF-C, sample synthesized using the chloride precursor; NNF-N, sample synthesized using the nitrate precursor.

**Table 2 | Measured fractions of β-NaNdF$_4$, NdF$_3$, and NaF phases after volatilization of the water of crystallization from the samples at 500 °C**

| Heated fluoride salt | Weight % of phases in the samples after drying at 500 °C | | |
|---|---|---|---|
| | NdF$_3$ | β-NaNdF$_4$ | NaF |
| NNF-A | 35 ± 1 | 64 ± 2 | 1 ± 0 |
| NNF-C | 54 ± 2 | 41 ± 2 | 5 ± 1 |
| NNF-N | 46 ± 1 | 48 ± 2 | 6 ± 1 |

volatilization of surface water, while the second peak at ~400–500 °C represents the volatilization of the crystallization water. These results corroborate the findings of the DSC studies that show a thermal event at ~400–500 °C (Fig. 2a), along with a mass loss observed in the TGA plot (Fig. 2b). The absence of HF evolution was confirmed by the flat line corresponding to mass number 20 in all the three fluorides. The absence of HF was further validated by heating the sample to 600 °C for 3 h (to ensure complete drying), while testing with a HF gas detector.

To investigate the phases in the NNF powder samples after the volatilization of water of crystallization, the as-synthesized samples were heat treated at 500 °C for 3 h and the powder X-ray diffraction (XRD) patterns were recorded (Fig. 4).

Three phases corresponding to β-NaNdF$_4$ (β-NNF) [space group: $P6_3/m$ (no. 176)], NdF$_3$ [space group: $P6_3/mcm$ (no. 193)], and NaF [space group: $Fm\bar{3}m$ (no. 225)] were observed. The phase fractions of the samples are presented in Table 2. The phase decomposition of the α-NaNdF$_4$ phase can be explained by the deviation from the single-phase region below 785 °C (Supplementary Fig. 8). Below this temperature, the composition corresponding to the α-NaNdF$_4$ phase falls in a two-phase region of β-NaNdF$_4$ and NdF$_3$[26]. Furthermore, below 732 °C, the β-NNF phase exists as a line compound, and any deviation in stoichiometry leads to a two-phase region of β-NaNdF$_4$ and NaF

phases (Supplementary Fig. 8). This explains the formation of three phases upon sample heating and is consistent with previous reports[26].

All the samples formed significant amounts of NdF$_3$ phase, in addition to the β-NaNdF$_4$ phase. NNF-A had the least amount of NaF while the amounts in NNA-C and NNA-N are comparable. The collected metal after calciothermic reduction is shown in Fig. 5a. The XRD of the reduced and polished metal in Fig. 5b shows peaks matching with Nd metal [space group: $P6_3/mmc$ (no. 194)]. The analysis of the Nd metal shows that it contains C:176 ppm, N: 52 ppm, O: 925 ppm, and S: 1 ppm. It is common for metals produced by calciothermic or electrolytic methods to undergo further purification which helps to reduce the impurities[30,31], such as those from the slag or the unreacted salts, etc. In the present work, the metal production yield was 70–80%, which is expected to increase at higher-scale operations.

TG-DSC measurement was performed on a sample of the metal, and the results are plotted in Fig. 6. The DSC plot depicts two endotherms at 852 °C and 1016 °C corresponding to a transformation of the unreacted fluoride sample from β-NaNdF$_4$ to α-NaNdF$_4$ and melting of metallic Nd, respectively. The TGA plot depicts little to no mass change, as expected for Nd metal.

## Methods

Neodymium acetate (99%) from Sigma Aldrich was used for this work. Neodymium chloride and neodymium nitrate were synthesized by dissolving industrial grade of neodymium hydroxide in hydrochloric and nitric acids, respectively, and drying at 80 °C. Neodymium chloride hexahydrate (10 mmol, 3.6 g) and neodymium nitrate hexahydrate (10 mmol, 4.4 g) were dissolved in 50 mL of water each. To this Nd salt solution, 2.1 g NaF in 50 ml water (50 mmol) were added, and the contents were stirred at 25 °C for 6 h. We found that a slightly lower ratio yielded no NaNdF$_4$, and a slightly higher ratio resulted in almost 100% NaNdF$_4$ but required even more excess NaF. The resulting NaNdF$_4$ precipitate was filtered, washed, and dried overnight at 80 °C in the air. To obtain dehydrated NaNdF$_4$ product, the sample was heat treated at 500 °C for 3 h in air.

Structural analysis of the NaNdF$_4$ product was performed by powder XRD using a Bruker D-8 X-ray diffractometer with Cu-Kα radiation. The XRD analysis was used to determine the phase constituents via the reference intensity ratio method with Match! Software (Version 3.15), and with Corundum as the reference. Simultaneous thermogravimetric analysis and differential scanning calorimetry (TGA-DSC) were performed using a STA449F1 system (Netzsch, Selb, Germany) with Al$_2$O$_3$ crucibles from 30 °C to 1050 °C. The temperature was scanned at 10 °C/min in ultra-high purity nitrogen gas flow (60 mL/min). Evolved gas analysis was performed using an Aeolos QMS 403D quadrupole mass spectrometer connected to the TGA furnace via a heated transfer line. Resublimed calcium prepared at Materials Preparation Center, Ames National Laboratory was used at a Ca: NaNdF$_4$ ratio of ~2.6 to reduce the NaNdF$_4$ to metals.

The reduction process was performed in a 0.75 × 5 × 1.2 (width × length × thickness) cubic inch tantalum crucible. A spun cap was welded at the bottom, and another loose spun cap was used as a lid. The tantalum crucible was placed in a quartz tube with a secondary quartz tube to retain the tantalum cap. The assembly was then suspended into an induction coil (Supplementary Fig. 1a), which was then heated from room temperature to 880 °C in 15 min. The reduction set-up was maintained under vacuum till 700 °C, and then backfilled with Ar to −100 mmHg (Supplementary Fig. 1b). The temperature was monitored with a two-color pyrometer until the coating obscured the view at ~880 °C (Supplementary Fig. 1c). The run continued for 23 min, following which an eruption of vapor and an induction heated plasma was observed. The induction power was terminated at this stage. The quartz tube was broken to bring out the sample. The contents were machined to remove the slag and tantalum to collect the neodymium metal (Supplementary Fig. 1d). The nitrogen and oxygen were analyzed

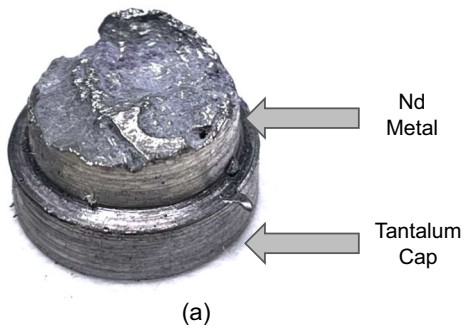

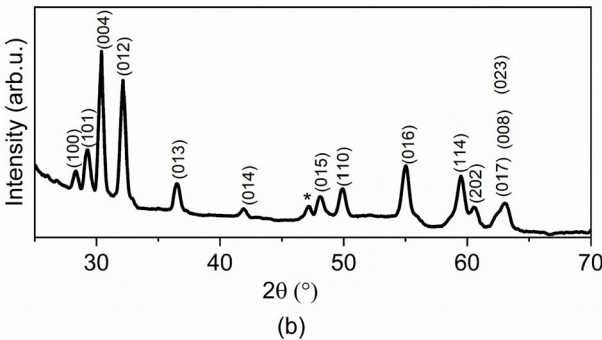

**Fig. 5 | Characterization of Nd metal produced by calciothermic reduction of sodium neodymium fluoride. a** Nd metal seated on tantalum cap after calciothermic reduction. The purple surface color is due to the unreacted material and slag on the metal, which was removed by mechanical polishing. The portions of the image corresponding to Nd metal and Ta cap are identified in the picture; **b** X-ray diffraction (XRD) pattern from the samples reduced to metal. All the peaks correspond to Nd metal [space group: $P6_3/mmc$ (no. 194)]. The asterisk represents $CaF_2$ phase from the slag.

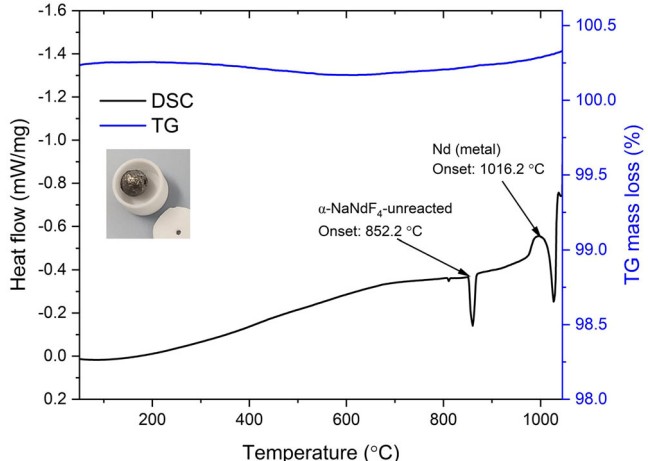

**Fig. 6 | TG-DSC of rare earth metal obtained from reducing rare-earth fluoride.** The thermogravimetric analysis (TGA) result (blue plot) shows little variation in the mass change over the temperature range. The differential scanning calorimetry (DSC) plot depicts two endothermic transformations likely corresponding to the melting of the unreacted α-NaNdF$_4$ phase (852.2 °C) and the melting of Nd metal (1016.2 °C). The inset is a spherical ball of Nd metal obtained after the TG-DSC experiment.

with LECO-ON836 using 0.1 g of sample in a graphite crucible under helium atmosphere. The carbon and sulfur were analyzed using LECO-CS844 using 0.5 g of sample in an alumina crucible under an $O_2$ atmosphere. The morphology of the fluoride samples was recorded using a FEI Teneo SEM.

The present study demonstrates the metallothermic reduction of fluoride salts to Nd metal, using NaNdF$_4$ synthesized via a facile route that excludes the use of HF. The fluoride salts were synthesized through room-temperature reactions involving only an aqueous solvent. The elimination of HF in synthesizing the fluorides improves the operational safety of the process. Three different rare earth salts have been utilized as starting materials to demonstrate the versatility of the process. Results show that only water evolved from the salts during the heating process.

## Data availability

Data sets generated during the current study are available from the corresponding author upon request. Determination of an acceptable request and subsequent data provision are subject to Ames National Laboratory approval.

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

## Acknowledgements

This work was supported by the Critical Materials Innovation Hub funded by the U.S. Department of Energy, Office of Energy Efficiency and Renewable Energy, Advanced Materials and Manufacturing Technologies Office. The work was performed at Ames National Laboratory, operated for the U.S. Department of Energy by Iowa State University of Science and Technology under Contract No. DE-AC02-07CH11358.

## Author contributions

A.K.: Data collection, Experimentation, Data analysis, Writing: original draft; H.P., T.R., and M.B.: Experimentation, Data analysis; D.P. and I.C.N.: Data analysis, Writing: review and editing, Supervision, Acquiring of funding.

## Competing interests

The authors declare the following competing interests. All authors are co-inventors on a related pending patent application entitled "preparation of rare earth metals with double salts". (Patent Application No. 20250075359). Applicant: Iowa State University Research Foundation, Inc. (Ames, IA).
