## [Transparent Peer Review file · Nature Communications]

Rare earth metals production using alternative feedstock that eliminates HF

Corresponding Author: Dr Ikenna Nlebedim

Version 0:

Reviewer comments:

Reviewer #1

(Remarks to the Author)

Comments to authors in the attached document

Reviewer #2

(Remarks to the Author)

The paper presents a novel approach to the production of rare earth (RE) metals, specifically focusing on the use of sodium rare earth fluoride (Na-RE-F) as an alternative to the traditionally employed rare earth fluoride (REF₃). While the paper presents a promising alternative for rare earth metal production, addressing the identified limitations and considering the implications of using various feedstocks would enhance its impact and applicability in the field. Further research is needed to explore the broader implications of using Na-RE-F across various rare earth metals, assess the scalability of the process, evaluate environmental impacts, and conduct comparative analyses with existing methods.

1. Comparative Analysis: There is a lack of detailed comparative analysis between the Na-RE-F method and existing methods using REF₃, particularly regarding cost-effectiveness and yield.

2. How suitable are the raw materials for the process?

1) Use of Non-Chemical Pure Reagents: If non-chemical pure reagents were utilized as feedstock, it would be essential to analyze the entire production process, including the synthesis of Na-RE-F. For instance, using industrial-grade neodymium salts could introduce variations in the purity and yield of the final product. The distribution of major elements would likely reflect the impurities present in the starting materials, potentially affecting the efficiency of the reduction process and the quality of the final neodymium metal.

2) Impact of Using Iron-Neodymium-Boron (NdFeB) Waste Magnets: Utilizing NdFeB waste magnets as a raw material could introduce additional elements such as iron (Fe) and boron (B) into the production process. The presence of iron may lead to contamination of the neodymium metal, affecting its magnetic properties and overall quality. Additionally, boron could complicate the metallothermic reduction process, potentially forming stable borides that hinder the reduction of neodymium. This could result in lower yields and necessitate further purification steps, thereby increasing production costs and complexity.

Reviewer #3

(Remarks to the Author)

The authors have attempted to synthesize NaNdF₄ through an aqueous chemistry which is further subjected to calciothermic reduction to produce neodymium metal. The proposed method reduces the dependence on HF, traditionally used for producing anhydrous NdF₃.

The manuscript is well written and provides an alternative cleaner approach for producing neodymium metal.

I have following suggestions/questions for the authors:

What was the basis for selection of the weights of salts used for synthesis of NaNdF₄? Is it based on stoichiometric calculations? The NaF dosage seems to be in excess. It will be worth adding a discussion on how this influence the precipitation chemistry and product.

The heating atmosphere for dehydrating Na-Nd-F product should be mentioned.

What was the oxygen content of the NaNdF₄ product and does it have any influence on high oxygen concentration in reduced metal? The concentration of oxygen in reduced metal is considerably higher than usual specification for use in magnet manufacturing.

What was the yield during the reduction process? It is mentioned that the reduced metal incorporated unreacted NaNdF₄: Does this imply low reduction potential of NaNdF₄ and therefore low yield and poor separation? What was the phase composition of slag obtained after reduction?

Version 1:

Reviewer comments:

Reviewer #2

(Remarks to the Author)

I am satisfied with the authors' responses to the previous comments and believe the revisions significantly strengthen the paper. The authors have effectively addressed my concerns, resulting in a well-structured and compelling manuscript. The innovative approach presented is a valuable contribution to the field, and I recommend acceptance.

Reviewer #3

(Remarks to the Author)

The authors have made the required corrections and added relevant data to the manuscript. I have no further comments.

Reviewer #4

(Remarks to the Author)

As we know, the current industrial process to obtain REM is electrolyzing from a molten fluoride using their oxides as feed. Fluoride or HF could bring some problems for post-processing process or product. However, fluoride-chloride mixed salt as a possible substitute can be solved to some extent in the recent research.

The authors proposed the method to produce neodymium metal by calciothermic reduction in this manuscript. The issue of continuity is something that needs to be focused on. Moreover, the physicochemical properties of NNF have been discussed in detail in references 26, 27 and 29, and lots of information can be obtained in Figure S8. Therefore, the novelty of the paper needs to be re-judged.

1. Considering the results presented in this paper, I am more interested in the thermodynamic data of some reactions, such as, reaction 1 and reactions in supporting information from 1 to 6. It's confusing that the number 5.93 in reaction 6.

2. It is difficult to wash and get a pure product from fluoride salt. So, my question is how did authors solve this problem perfectly?

Version 2:

Reviewer comments:

Reviewer #4

(Remarks to the Author)

The authors sincerely appreciate the effort invested in reviewing our manuscript. We have revised the manuscript in accordance with the reviewers' comments, which has greatly improved it. Please note that changes made to address the reviewers' comments are shown in red font in the manuscript and supplementary information files, along with general edits to enhance the manuscript. In this document, our responses are in blue fonts.

Reviewer #1 (Remarks to the Author):

The manuscript gathers the results obtained after metallothermic reduction of NaNdF_4 salt synthesized via simple aqueous routes using several neodymium salts (acetate, chloride and nitrate), and without involving the use of HF. The authors proposed the use of the obtained NNF salts to produce neodymium metal by calciothermic reduction, showing the results obtained in a bench-scale trial. The proposed process is presented by the authors as a possible substitute of the current industrial process to obtain Nd metal, i.e., electrolysis from a molten fluoride using Nd oxide as feed. This seems hard to believe, though the manuscript has a certain value from a fundamental scientific point of view. The manuscript has currently a somewhat pure quality and could eventually be published after major revision. The following must be corrected and clarified:

General Comments on the reported work

1. Calciothermic reduction is a process to manufacture rare earth (RE) metals, using REF_3 as raw material. This is the current industrial process to obtain Samarium metal. However, electrolysis is the most common method to produce RE metals that are needed for the manufacture of Nd-based permanent magnets, i.e., Nd metal, Nd/Pr mischmetal, as well as Dy-Fe and Tb-Fe alloys. The electrolysis process is carried out in a REF_3 -based electrolyte (typically $\text{REF}_3\text{-LiF}$) using the RE-oxide as feed. In the electrolysis process, the REF_3 is not the feed, but the RE_2O_3 , the REF_3 being part of the electrolyte, only, and it is not "consumed" during the process, apart from possible evaporation losses. This must be corrected in the manuscript (Introduction Chapter).

Reply: We agree with the reviewer and have edited the associated sentence to reflect the recommendation.

See page 2 of the manuscript for the relevant edits.

2. The authors explained the results obtained in QMID, by volatilization of H_2O , both from the adsorbed water and the water of crystallization of the synthesized salt samples. It is hard to believe the absence of HF evolution (flat line observed), considering the significant presence of NdF_3 in the NaNdF_4 samples obtained from the neodymium chloride and nitrate salts (cf. Table 1), and its reactivity towards water at high temperatures.

Reply: This is an important comment by the reviewer which also strengthens the utility of our approach. It is interesting to observe that the same samples referenced by the reviewer also had the least amount of mass loss during drying (<0.5%) as shown in Figure 2b. In our work, the sample derived from acetate had 100% NaNdF_4 but more mass loss during drying. This suggests that NaNdF_4 likely contains relatively more water than NdF_3 .

To further investigate, we have performed additional work in which we dried 100g of Na-Nd-F sample at 600 °C for 3h. A higher temperature was used to ensure complete drying of the

sample. No HF was detected even when tested with a HF gas detector as shown below. No changes have been made to the manuscript, given that the results are consistent.

See page 6 of the manuscript for the relevant edits.

The HF concentration (ppm) was monitored with a hand-held HF detector during the annealing of a 100g sample of NaNdF_4 in air, over a temperature range of 100 to 600°C. The inset shows 0.0 ppm reading on the HF detector at 600 °C.

3. The results obtained by the authors are explained by the presence of water of crystallization in the synthesized NaNdF_4 samples. However, any of the equations showed this fact, and should be corrected, i.e., Equation (1) in the manuscript, and Equation (5) in the Supplementary Information document.

Reply: Thank you for pointing out this omission which has now been addressed.

4. According to the results showed in the XRD diffractograms in Figure 1, the synthesized compound is $\alpha\text{-NaNdF}_4$. However, the phase diagram showed in Figure S8 shows that $\beta\text{-NaNdF}_4$ (hexagonal phase) is stable at low temperatures, and $\alpha\text{-NaNdF}_4$ (cubic phase) at higher. Phase transformation temperature is, according to the authors, ~ 800 °C (cf. page 5). This must be clarified.

Reply: We remark that the phase diagram shown in Figure S8 is between NaF and NdF_3 , instead of NaF and Nd-salts (i.e., Nd-acetate, Nd-chloride and Nd-nitrate), which are used for our work. Nevertheless, the referee made an interesting remark which was what we originally expected. However, others who used Nd-salts and NaF as precursors have also reported the formation of $\alpha\text{-NaNdF}_4$ (cubic phase) between 25-200 °C [R1, R2]. Interestingly, upon heating to 500 °C for 3hr and cooling, instead of the $\alpha\text{-NaNdF}_4$, $\beta\text{-NaNdF}_4$ was observed, suggesting that non-equilibrium conditions that may occur during co-precipitation could lead to the stabilization of the $\alpha\text{-NaNdF}_4$ at low temperatures.

- R1. Yang, S., Jayanthi, K., Anderko, A., Riman, R. E. & Navrotsky, A. Thermochemical Investigation of the Stability and Conversion of Nanocrystalline and High-Temperature Phases in Sodium Neodymium Fluorides. *Chemistry of Materials* 33, 9571–9579 (2021).
- R2. Kuznetsov, S. V. *et al.* Phase formation in LaF₃–NaGdF₄, NaGdF₄–NaLuF₄, and NaLuF₄–NaYF₄ systems: Synthesis of powders by co-precipitation from aqueous solutions. *J Fluor Chem* 161, 95–101 (2014).
5. The results obtained in the synthesis of NaNdF₄ samples from the different Nd-salts as precursors were analyzed and quantified, in terms of NaNdF₄ and NdF₃ content (Table 1). The content of the initial substances/reactants, i.e., Nd-salts and NaF, is not given, then assuming that the conversion to NaNdF₄ is complete. However, DSC data reveals signals corresponding to the eutectic mixtures of β–NaNdF₄ and NaF (broad peaks obtained in the range 600–700 °C, obtained upon heating, cf. text in page 5 and Figure 2).

Reply: The eutectic mixture of β–NaNdF₄ and NaF observed between 600 – 700 °C resulted from heating the materials, rather than being present in the synthesized NaNdF₄ sample. Hence, we believe that the reaction was complete. However, we agree with the reviewer that the descriptions in Table 1 needs improvement and have done so.

Other Comments

- Abbreviations must be explained the first time they are shown in the text: The NaNdF₄ compound it is referred to as NNF at a certain point in the manuscript (from legend in Figure 1 (page 4) and in the following text (from page 5). This must be clarified in the text, as well as in Table 1, where the NaNdF₄ compounds synthesized from the different Nd-salts are named, e.g. NNF-A, NNF-C, NNF-N.
Reply: Thank you for this guidance. It is now addressed.
- XRD, from legend in Figure 1, page 4, and text in page 3.
Reply: It is now addressed.
- SEM, from page 3.
Reply: It has been addressed.
- DSC, differential scanning calorimetry, from page 5.
Reply: It is now addressed.
- TGA, thermal gravimetric analysis, from page 5.
Reply: It has been addressed.
- The chemical formula of the compounds must be expressed correctly throughout the manuscript, e.g., RE must be changed to RE-fluoride(s) or RE₃F₉, and NaNdF₄ compound must be changed to NaNdF₄, or its abbreviation NNF.
Reply: The changes have been incorporated in multiple pages of the revised manuscript.
- Some explanations of the Figures are included in the text, rather than in the legends. This is the case of, e.g., Figure 4 where the explanation of the different curves corresponding to the

different Nd salts is given in the text, rather than in the legends. Authors must revise the whole manuscript/legend of Figures, and correct when needed.

Moreover, the legends must be revised and include more information when needed, e.g. Figure 2, explain what is (a) and (b); Figure 5(b) does not specify the correspondence of the different peaks obtained in the diffractogram, but only the crystalline planes...

Reply: Thank you. All the legends have been modified in the revised manuscript.

- Introduction Chapter, page 2. When explaining the current electrolysis process to obtain Nd, either from RE-chloride or RE-fluoride based melts, the feedstock must be better specified. **Instead of:** “*The electrolyte for the molten-salt bath can be chloride or fluoride-based, and the feedstock can be RE-oxides or RE-halides*”, **it should be stated:** “*The electrolyte for the molten-salt bath can be chloride- or fluoride-based, and the feedstock can be RE-chloride or RE-oxide, respectively*”.

Reply: Thank you. The suggested modification has been incorporated.

- Table 2 legend states that both the “measured phase fraction” and “lattice parameters” are shown in the Table. However, only the “measured phase fraction” (wt%) of the different compounds are provided, not the lattice parameters.

Reply: Thanks for identifying the error which is now corrected.

- In the description of the Methods, page 11, it is stated that TGA-DSC tests were carried out in Al₂O₃ crucibles in a temperature range of 30-1000 °C. This cannot be correct in the case of the test carried out in the case of Nd metal obtained after calciothermic reduction, as the temperature needed to melt Nd should be 1016.2 °C or higher. Moreover, the alumina crucible used during the tests looks extremely clean, considering the high reactivity of molten Nd towards any ceramic material (cf. picture insert in Figure 6).

Reply: The TG-DSC was measured in the temperature window of 30-1050 °C. The change has been incorporated into the revised manuscript on Page 11.

The text corresponding to the inset of Figure 6 has been revised on Page 9 of the revised manuscript.

Reviewer #2 (Remarks to the Author):

1. The paper presents a novel approach to the production of rare earth (RE) metals, specifically focusing on the use of sodium rare earth fluoride (Na-RE-F) as an alternative to the traditionally employed rare earth fluoride (REF₃). While the paper presents a promising alternative for rare earth metal production, addressing the identified limitations and considering the implications of using various feedstocks would enhance its impact and applicability in the field. Further research is needed to explore the broader implications of using Na-RE-F across various rare earth metals, assess the scalability of the process, evaluate environmental impacts, and conduct comparative analyses with existing methods.

Comparative Analysis: There is a lack of detailed comparative analysis between the Na-RE-F method and existing methods using REF₃, particularly regarding cost-effectiveness and yield.

Reply: We thank the reviewer for the insightful and constructive feedback and appreciate the recognition of the novel approach we have presented in our manuscript. Also, we acknowledge the importance of addressing the identified limitations and considering the implications of using various feedstocks to enhance the impact and applicability of our work. Your suggestions for further research into the broader implications of Na-RE-F, scalability, environmental impacts, and comparative analyses with existing methods are valuable and will certainly guide our future research efforts.

To ensure a balanced study, we aim to undertake future work in which REF₃ and Na-RE-F will be systematically compared, including the processing conditions, yield, economic and environmental impacts, etc. At this stage, we aim to focus on the current scope of our study, but we look forward to exploring these important aspects in subsequent research projects.

2. How suitable are the raw materials for the process?

a) Use of Non-Chemical Pure Reagents: If non-chemical pure reagents were utilized as feedstock, it would be essential to analyze the entire production process, including the synthesis of Na-RE-F. For instance, using industrial-grade neodymium salts could introduce variations in the purity and yield of the final product. The distribution of major elements would likely reflect the impurities present in the starting materials, potentially affecting the efficiency of the reduction process and the quality of the final neodymium metal.

Reply: This is an excellent comment in view of future large-scale commercial deployment. Indeed, a critical aspect of materials production is the control of the composition of the feedstock. We expect such to be applicable to our novel process. To ensure that such is achievable, in the present work we used industrial-grade neodymium hydroxide (98-99% pure) to prepare NdCl₃.6H₂O and Nd(NO₃)₃.6H₂O which was, in turn, converted to α -NNF.

See page 9 of the manuscript for the relevant edits.

b) Impact of Using Iron-Neodymium-Boron (NdFeB) Waste Magnets: Utilizing NdFeB waste magnets as a raw material could introduce additional elements such as iron (Fe) and boron (B) into the production process. The presence of iron may lead to contamination of the neodymium metal, affecting its magnetic properties and overall quality. Additionally, boron could complicate the metallothermic reduction process, potentially forming stable borides that hinder the reduction of neodymium. This could result in lower yields and necessitate further purification steps, thereby increasing production costs and complexity.

Reply: The process we reported would not directly utilize waste NdFeB magnet. The reason for such is as the reviewer stated. Nevertheless, RE₂O₃ from a recycling process, such as the acid-free dissolution recycling (ADR), that can achieve 99.9% pure RE₂O₃ can be used. Moreover, it is common to perform additional purification steps after metallothermic or calciothermic process.

See page 8 of the manuscript for the relevant edits.

Reviewer #3 (Remarks to the Author):

1. The authors have attempts to synthesize NaNdF_4 through an aqueous chemistry which is further subjected to calciothermic reduction to produce neodymium metal. The proposed method reduces the dependence on HF, traditionally used for producing anhydrous NdF_3 . The manuscript is well written and provides an alternative cleaner approach for producing neodymium metal. I have following suggestions/questions for the authors:

Reply: We thank the reviewer for the kind comments regarding the novelty of our approach.

2. What was the basis for selection of the weights of salts used for synthesis of NaNdF_4 ? Is it based on stoichiometric calculations? The NaF dosage seems to be in excess. It will be worth adding a discussion on how this influences the precipitation chemistry and product.

Reply: The ratio of Nd-salts to NaF was selected considering previous research. We also validated those reports in our own research. Moreover, to respond to the reviewer's comments, we have performed additional experimental work in that respect. We used varying ratios of the materials for synthesis, and performed XRD to determine the phase fractions. In brief, 1:4 ratio yielded almost 0% NaNdF_4 and 1:6 ratio yielded 99% NaNdF_4 but required even more excess NaF. Hence, 1:5 ratio maximizes the amount of NaNdF_4 without requiring additional excess NaF. A statement that reflects this has been included in the manuscript with a reference.

See pages 9 and 10 of the manuscript for the relevant edits.

3. The heating atmosphere for dehydrating Na-Nd-F product should be mentioned.

Reply: Thank you. The details have been added on Page 11 of the revised manuscript.

4. What was the oxygen content of the NaNdF_4 product, and does it have any influence on high oxygen concentration in reduced metal? The concentration of oxygen in reduced metal is considerably higher than usual specification for use in magnet manufacturing.

Reply: In future large-scale deployments of this process, materials handling would be simplified if drying occurs in air, as demonstrated in our current work. Therefore, measuring oxygen content after heating in air would not be necessary. In practice, an excess of calcium is used to ensure complete reduction. Additionally, the Nd metal obtained here would typically undergo further purification through processes such as vacuum induction melting & casting, or distillation. These subsequent steps reduce the oxygen content.

5. What was the yield during the reduction process? It is mentioned that the reduced metal incorporated unreacted NaNdF_4 : Does this imply low reduction potential of NaNdF_4 and therefore low yield and poor separation? What was the phase composition of slag obtained after reduction?

Reply: This initial work was conducted at the laboratory scale, achieving yields in the range of 70–80%. However, based on established calciothermic reduction processes, we expect that scaling up to industrial levels will result in significantly improved yields, typically exceeding 90%. Therefore, the current yield should be considered reflective of the constraints at the lab scale.

In large-scale operations, it is standard practice to further purify metals produced via calciothermic and electrolytic processes. Techniques such as induction melting are commonly employed to enhance the separation of metal from slag, resulting in higher purity. The slag

composition varies across its layers, predominantly consisting of CaF_2 and NaF , along with the reduced Nd metal and minor traces of CaO and NaNdF_4 .

See page 8 of the manuscript for the relevant edits.

The authors sincerely appreciate the effort invested in reviewing our manuscript. We have provided responses to the reviewers' comments and revised the manuscript accordingly, which has improved it. Please note that changes made to address the reviewers' comments are shown in red font in the manuscript. In this document, our responses are in blue fonts.

Reviewer #3 (Remarks to the Author):

1. The authors have made the required corrections and added relevant data to the manuscript. I have no further comments.

Response: We sincerely appreciate the referee's thoughtful remarks regarding our revisions to the manuscript. As mentioned in our previous submission, the referee's guidance is invaluable in enhancing the quality of our work.

Reviewer #4 (Remarks to the Author):

1. As we know, the current industrial process to obtain REM is electrolyzing from a molten fluoride using their oxides as feed. Fluoride or HF could bring some problems for post-processing process or product. However, fluoride-chloride mixed salt as a possible substitute can be solved to some extent in the recent research.

Response: This is an interesting perspective from the reviewer which could be a possible approach to addressing challenges to molten salt electrolysis processes for rare earth metals production.

2. The authors proposed the method to produce neodymium metal by calciothermic reduction in this manuscript. The issue of continuity is something that needs to be focused on. Moreover, the physicochemical properties of NNF have been discussed in detail in references 26, 27 and 29, and lots of information can be obtained in Figure S8. Therefore, the novelty of the paper needs to be re-judged.

Response: Although NNF has been previously studied, our work presents a novel contribution as the first demonstration of its use in rare earth metal production. The novelty of our work is in leveraging NNF to meet the chemical requirements of the traditional approach for rare earth metal production including: fluoride ion availability, lower reduction temperature and compatibility with rare earth reduction processes as herein demonstrated. Beyond its novelty, this approach offers clear practical benefits, particularly in metallothermic reduction processes. The adoption of NNF as an alternative fluoride salt for rare earth metal production is not merely a matter of convenience but a significant safety and sustainability advancement. It offers a practical pathway for reducing the environmental and operational hazards associated with the use of conventional fluoride salts. Implementing NNF represents a strategic shift toward safer, more sustainable, and regulation-compliant rare earth metal production, without compromising process efficiency or product quality. To this end, we have added the following text to the manuscript:

“Unlike conventional RE-fluoride reduction, NNF not only ensures fluoride ion availability but also represents a novel and strategic shift toward safer, more sustainable, and regulation-compliant rare earth metal production.”

3. Considering the results presented in this paper, I am more interested in the thermodynamic data of some reactions, such as, reaction 1 and reactions in supporting information from 1 to 6. It's confusing that the number 5.93 in reaction 6.

We appreciate the reviewer's interest in the thermodynamic data for reactions leading to NNF from the rare earth salts. While our primary focus is on demonstrating the suitability of NNF for rare earth metal production rather than the thermodynamics of its formation, we acknowledge the limited availability of such data. To address this, we have now cited a recent study that provides an investigation of the heat capacity and thermodynamic properties of Na-REE-F compounds, including NNF, from 1.8 to 300 K. This reference strengthens the context of our work and offers valuable insights for readers interested in the thermodynamic aspects of these materials. See page 3 of the manuscript in which this statement was added:

“A recent study by Gibson et al. reported a negative Gibb's energy for NNF, indicating its thermodynamic stability at room temperature (298.15 K)”.

Regarding reaction 6, we acknowledge that whole numbers are generally preferred in balanced chemical equations. However, representing certain coefficients in decimal form provides valuable insight into structural order in solids and crystallization water content. This reaction is conceptually similar to Equation 1 (in the manuscript) and allows for a direct comparison across different starting salts:

4. It is difficult to wash and get a pure product from fluoride salt. So, my question is how did authors solve this problem perfectly?

The reviewer is correct that washing is necessary to obtain a higher-purity fluoride product. However, in our process, purification was not significantly challenging, as the primary byproducts are highly soluble in water (see table below), in comparison to the fluoride salts (~0.0016 g/L). This high solubility differences allowed for effective washing, as confirmed by our XRD results.

Table R1: Solubility of different salts in water at 298 K.

#	Chemical name	Solubility (g/L)	References
1.	Neodymium Acetate Hydrate	262	[1]
2.	Neodymium Chloride Hexahydrate	1000	[2]
3.	Neodymium Nitrate Hexahydrate	Highly soluble	[3]
4.	Sodium Chloride	360	[4]
5.	Sodium Acetate Trihydrate	460	[5]
6.	Sodium Nitrate	912	[6]

That said, we acknowledge that large-scale production may introduce greater challenges in washing the fluoride salt, particularly given the need to optimize water usage. However, potential residual impurities, such as NaF (42 g/L solubility in water) [7], are unlikely to impact the effectiveness of rare-earth metal production, which remains the primary focus of this study. It is

common for metals produced by calciothermic or electrolytic methods to undergo further purification which helps to reduce the impurities, such as those from the slag or the unreacted salts, etc. Techniques such as induction melting are routinely employed to facilitate the separation of metal from slag to improve purity.

References

1. Seidell, Atherton; Linke, William F. (1952). Solubilities of Inorganic and Organic Compounds. Van Nostrand.
2. Haynes, William M., ed. (2016). CRC Handbook of Chemistry and Physics (97th ed.). CRC Press. p. 4.75. ISBN 9781498754293.
3. Templeton, Charles C. (1949). The Distribution of Rare Earth Nitrates between Water and n-Hexyl Alcohol at 25°, Journal of the American Chemical Society 71(6), 2187-2190.
4. Haynes, William M., ed. (2011). CRC Handbook of Chemistry and Physics (92nd ed.). CRC Press. p. 4.89.
5. Seidell, Atherton; Linke, William F. (1952). Solubilities of Inorganic and Organic Compounds. Van Nostrand.
6. Haynes, William M. (2016-06-22). CRC Handbook of Chemistry and Physics. CRC Press. ISBN 978-1-4987-5429-3.
7. Haynes, William M., ed. (2011). CRC Handbook of Chemistry and Physics (92nd ed.). CRC Press. p. 5.194. ISBN 978-1-4398-5511-9.

Reviewer report. Manuscript Number NCOMMS-24-55103

The manuscript gathers the results obtained after metallothermic reduction of NaNdF_4 salt synthesized via simple aqueous routes using several neodymium salts (acetate, chloride and nitrate), and without involving the use of HF. The authors proposed the use of the obtained NNF salts to produce neodymium metal by calciothermic reduction, showing the results obtained in a bench-scale trial.

The proposed process is presented by the authors as a possible substitute of the current industrial process to obtain Nd metal, i.e., electrolysis from a molten fluoride using Nd oxide as feed. This seems hard to believe, though the manuscript has a certain value from a fundamental scientific point of view.

The manuscript has currently a somewhat pure quality and could eventually be published after major revision. The following must be corrected and clarified:

General Comments on the reported work

- Calciothermic reduction is a process to manufacture rare earth (RE) metals, using REF_3 as raw material. This is the current industrial process to obtain Samarium metal. However, electrolysis is the most common method to produce RE metals that are needed for the manufacture of Nd-based permanent magnets, i.e., Nd metal, Nd/Pr mischmetal, as well as Dy-Fe and Tb-Fe alloys. The electrolysis process is carried out in a REF_3 -based electrolyte (typically $\text{REF}_3\text{-LiF}$) using the RE-oxide as feed. In the electrolysis process, the REF_3 is not the feed, but the RE_2O_3 , the REF_3 being part of the electrolyte, only, and it is not “consumed” during the process, apart from possible evaporation losses. This must be corrected in the manuscript (Introduction Chapter).
- The authors explained the results obtained in QMID, by volatilization of H_2O , both from the adsorbed water and the water of crystallization of the synthesized salt samples. It is hard to believe the absence of HF evolution (flat line observed), considering the significant presence of NdF_3 in the NaNdF_4 samples obtained from the neodymium chloride and nitrate salts (cf. Table 1), and its reactivity towards water at high temperatures.
- The results obtained by the authors are explained by the presence of water of crystallization in the synthesized NaNdF_4 samples. However, any of the equations showed this fact, and should be corrected, i.e., Equation (1) in the manuscript, and Equation (5) in the Supplementary Information document.
- According to the results showed in the XRD diffractograms in Figure 1, the synthesized compound is $\alpha\text{-NaNdF}_4$. However, the phase diagram showed in Figure S8 shows that $\beta\text{-NaNdF}_4$ (hexagonal phase) is stable at low temperatures, and $\alpha\text{-NaNdF}_4$ (cubic phase) at higher. Phase transformation temperature is, according to the authors, ~ 800 °C (cf. page 5). This must be clarified.
- The results obtained in the synthesis of NaNdF_4 samples from the different Nd-salts as precursors where analyzed and quantified, in terms of NaNdF_4 and NdF_3 content (cf.

Table 1). The content of the initial substances/reactants, i.e., Nd-salts and NaF, is not given, then assuming that the conversion to NaNdF_4 is complete. However, DSC data reveals signals corresponding to the eutectic mixtures of β - NaNdF_4 and NaF (broad peaks obtained in the range 600-700 °C, obtained upon heating, cf. text in page 5 and Figure 2).

Other Comments

- Abbreviations must be explained the first time they are shown in the text:
 - The NaNdF_4 compound it is referred to as NNF at a certain point in the manuscript (from legend in Figure 1 (page 4) and in the following text (from page 5). This must be clarified in the text, as well as in Table 1, where the NaNdF_4 compounds synthesized from the different Nd-salts are named, e.g. NNF-A, NNF-C, NNF-N.
 - XRD, from legend in Figure 1, page 4, and text in page 3.
 - SEM, from page 3.
 - DSC, differential scanning calorimetry, from page 5.
 - TGA, thermal gravimetric analysis, from page 5.
- The chemical formula of the compounds must be expressed correctly throughout the manuscript, e.g., RE must be changed to RE-fluoride(s) or REF_3 , and NaNdF compound must be changed to NaNdF_4 , or its abbreviation NNF.
- Some explanations of the Figures are included in the text, rather than in the legends. This is the case of, e.g., Figure 4 where the explanation of the different curves corresponding to the different Nd salts is given in the text, rather than in the legends. Authors must revise the whole manuscript/legend of Figures, and correct when needed.

Moreover, the legends must be revised and include more information when needed, e.g. Figure 2, explain what is (a) and (b); Figure 5(b) does not specify the correspondence of the different peaks obtained in the diffractogram, but only the crystalline planes...
- Introduction Chapter, page 2. When explaining the current electrolysis process to obtain Nd, either from RE-chloride or RE-fluoride based melts, the feedstock must be better specified. **Instead of:** *“The electrolyte for the molten-salt bath can be chloride- or fluoride-based, and the feedstock can be RE-oxides or RE-halides”, it should be stated:* *“The electrolyte for the molten-salt bath can be chloride- or fluoride-based, and the feedstock can be RE-chloride or RE-oxide, respectively”.*
- Table 2 legend states that both the “measured phase fraction” and “lattice parameters” are shown in the Table. However, only the “measured phase fraction” (wt%) of the different compounds are provided, not the lattice parameters.
- In the description of the Methods, page 11, it is stated that TGA-DSC tests were carried out in Al_2O_3 crucibles in a temperature range of 30-1000 °C. This cannot be correct in the case of the test carried out in the case of Nd metal obtained after calciothermic

reduction, as the temperature needed to melt Nd should be 1016.2 °C or higher. Moreover, the alumina crucible used during the tests looks extremely clean, considering the high reactivity of molten Nd towards any ceramic material (cf. picture insert in Figure 6).